# Increased Diagnostic Accuracy of Adnexal Tumors with A Combination of Established Algorithms and Biomarkers

**DOI:** 10.3390/jcm9020299

**Published:** 2020-01-21

**Authors:** Maria Lycke, Benjamin Ulfenborg, Björg Kristjansdottir, Karin Sundfeldt

**Affiliations:** 1Department of Obstetrics and Gynecology, Institute of Clinical Science, Sahlgrenska Academy, University of Gothenburg and Region Västra Götaland, Sahlgrenska University Hospital, 41345 Gothenburg, Sweden; bjorg_konsult@hotmail.com (B.K.); karin.sundfeldt@gu.se (K.S.); 2Systems biology research center, School of Bioscience, University of Skövde, 54128 Skövde, Sweden; benjamin.ulfenborg@his.se

**Keywords:** ovarian neoplasm, diagnosis, algorithms, RMI, ROMA, CA125, HE4

## Abstract

Ovarian cancer is the most lethal gynecologic cancer. Pre-diagnostic testing lacks sensitivity and specificity, and surgery is often the only way to secure the diagnosis. Exploring new biomarkers is of great importance, but the rationale of combining validated well-established biomarkers and algorithms could be a more effective way forward. We hypothesized that we can improve differential diagnostics and reduce false positives by combining (a) risk of malignancy index (RMI) with serum HE4, (b) risk of ovarian malignancy algorithm (ROMA) with a transvaginal ultrasound score or (c) adding HE4 to CA125 in a simple algorithm. With logistic regression modeling, new algorithms were explored and validated using leave-one-out cross validation. The analyses were performed in an existing cohort prospectively collected prior to surgery, 2013–2016. A total of 445 benign tumors and 135 ovarian cancers were included. All presented models improved specificity at cut-off compared to the original algorithm, and goodness of fit was significant (*p* < 0.001). Our findings confirm that HE4 is a marker that improves specificity without hampering sensitivity or diagnostic accuracy in adnexal tumors. We provide in this study “easy-to-use” algorithms that could aid in the triage of women to the most appropriate level of care when presenting with an unknown ovarian cyst or suspicious ovarian cancer.

## 1. Introduction

Most of the adnexal masses found by transvaginal ultrasound (TVU) or other imaging techniques such as computer tomography (CT) are not cancer. To be certain, surgical resection of the ovaries or the ovarian tumor is advised, often together with a complete staging that includes hysterectomy, omentectomy, lymph node sampling, and random peritoneal biopsies [1]. The established biomarker serum cancer antigen 125 (CA125) and the algorithms such as risk of malignancy index (RMI) and risk of ovarian malignancy algorithm (ROMA), which also include serum human epididymis 4 (HE4), have increased our ability to triage women to the correct level of care. Still, over fifty percent of the surgically resected specimens are benign, even in patients treated at highly specialized gynecologic cancer centers [2,3,4]. To decrease mortality in epithelial ovarian cancer (EOC), we need biomarkers for screening or early detection, and patients with an advanced disease should undergo optimal treatment [5,6,7]. Furthermore, women with benign disease should not impose on the limited capacity of the tertiary center.

EOC is the most lethal gynecologic malignancy and the fifth leading cause of death due to cancer among women worldwide [8]. It is often diagnosed at an advanced stage, which is partly due to variable and unspecific clinical symptoms and a lack of accurate diagnostic tools at early stages. TVU is widely used in open clinics to assess whether a pelvic tumor is malignant or benign in the preoperative workup. The TVU score is one of the parameters in the RMI, and is one of the first and most widely implemented algorithms for the triage in a tertiary center [9,10]. HE4 is a secreted glycoprotein discovered in 1991, found overexpressed in EOC 1999, and shortly after Hellstrom et al. suggested HE4 as a biomarker for EOC [11,12,13]. Importantly, unlike CA125, HE4 is not elevated in several benign conditions, which increases its specificity [2,14,15]. After the discovery of HE4, a TVU-independent triage by ROMA was described combining CA125, HE4, and menopausal status [16]. In larger studies, RMI and ROMA are equally good but have individual differences [1].

Serum CA125, a parameter in RMI and ROMA, is currently one of the best biomarkers to detect the most common subtype of EOC, high grade serous ovarian cancer (HGSC), in women above 50 years of age [17]. CA125 has several disadvantages when used as a biomarker in younger women since it is elevated due to inflammation, pregnancy, and endometriosis [18]. The TVU score, CA125, and menopausal status are combined in RMI [9,19]. In recent years HE4, approved by the US Food and Drug Administration (FDA), has been shown to be a good complement to CA125 in the differential diagnostics between benign and malignant ovarian tumors.

To improve specificity for TVU a doppler ultrasound and morphology index have been tested in different models, such as logistic regression models 1 and 2 (LR1 and LR2), simple rules, the ADNEX model, and simple rules risk score from the International Ovarian Tumor Analysis (IOTA) group [20,21]. The majority of these models require considerable experience in TVU, which might not be provided in a general gynecologic setting [22]. In addition to modified TVU algorithms (IOTA), new multi-biomarker assays and algorithms are now on the market along with several less validated early diagnostic biomarkers found through high-throughput technologies like mass spectrometry [16,23,24,25,26,27,28,29,30,31], recently reviewed [32]. Exploring novel biomarkers is of great importance, but perhaps the combination of well-established biomarkers and algorithms could be a more effective way forward than the discovery of new ones.

In the present study, we hypothesized that we could improve the specificity and reduce false positives by combining HE4 with RMI, TVU with ROMA, or by adding HE4 to CA125 in simple algorithms. We used data from a prior study by Lycke et al. (2018), where the TVU score according to RMI, menopausal status, and serum CA125 and HE4 were prospectively and continuously collected in a multicenter study, 2013–2016, from women scheduled for surgery due to an unknown pelvic mass [2]. We have assessed the three novel algorithms’ clinical performance by comparing specificity at target sensitivity and sensitivity at fixed specificity (75%) against the original models.

## 2. Materials and Methods

A large prospective multicenter clinical trial was conducted in Western Sweden (6 hospitals) between 09-2013 and 02-2016 to evaluate the diagnostic performance of HE4, CA125, RMI, and ROMA according to clinically established cut-offs in an unselected population of 638 women with an unknown pelvic mass (ClinicalTrials.gov NCT03193671) [2]. Patients included were assessed with transvaginal ultrasound (TVU) by both highly specialized sonographers and doctors undergoing training according to RMI criteria [19]. Surgically excised specimens were examined by an experienced pathologist for gynecological diseases. The tumors were categorized into histopathology, grade, and stage (I–V) according to The International Federation of Gynecology and Obstetrics (FIGO) 2014 standards, and divided into Type I and Type II EOC according to Kurman et al. [33] (Table 1). The local ethics committee of Gothenburg University approved the study protocol for each recruitment center (Ref 139-13).

For the present study, a total of 638 women, aged 18–87 were included (Table 1; Figure 1). Postmenopausal (Post-M) status was defined as one year of amenorrhea or being > 50 years of age with previous hysterectomy, the remaining subjects were defined as premenopausal (Pre-M). Sampling, storage, and shipping to Sahlgrenska University hospital were conducted according to established routines at authorized laboratories as previously described [2]. The assays for serum HE4 and CA125 were performed on coded samples using the Elecsys HE4 and Elecsys CA125 II with the electrochemiluminescence (ECLIA) technique (Cobas 8000, Roche Diagnostics Scandinavia, Stockholm, Sweden). Recommended cut-offs according to the manufacturer were used, CA125 > 35 U/mL, HE4 Pre-M > 70 pmol/L, and HE4 Post-M > 140 pmol/L [34]. Benign cases above cut-off were defined as false positives and malignant cases below cut-off as false negatives.

RMI score was calculated according to Tingulstad et al. [19]. RMI = U × M × serum CA125, where U = Ultrasound of 1 for imaging score 0–1 and U = 3 for an imaging score of 2–5; M = Menopausal status, M = 1 if Pre-M and M = 3 if Post-M. RMI > 200 was used as the cut-off.

The ROMA score was calculated according to Moore et al. [16]. The Pre-M predictive index (PI) = −12.0 + 2.38 × LN(HE4) + 0.0626 × LN(CA125); the Post-M PI = −8.09 + 1.04 × LN(HE4) + 0.732 × LN(CA125). The predicted probability was calculated as ROMA% = exp. (PI)/(1 + exp. (PI)) [16]. Cut-off levels for Pre-M ≥ 11.4% and Post-M ≥ 29.9% were used.

Statistical calculations were carried out in R, version 3.6.0 (R Core Team 2019), based on the patient cohort including benign tumors (*n* = 445) and epithelial ovarian cancers (*n* = 135). Subgroup analyses were performed on early stage tumors (FIGO I + II; *n* = 52), late stage tumors (FIGO III + IV; *n* = 83), borderline type ovarian tumors (BOT; *n* = 31), and metastasis to the ovary (*n* = 27; Table 1). Three logistic regression models were fitted using the following predictors: RMI + HE4, ROMA + TVU, and CA125 + HE4. RMI, HE4, and CA125 were log2 transformed prior to analysis. A binary response variable was used where benign and ovarian cancer samples were coded as 0 and 1, respectively. The models were compared to their respective baseline models (RMI, ROMA, CA125) to assess goodness of fit with the likelihood ratio test. To assess the clinical performance, sensitivity (SN), specificity (SP), and area under the curve (AUC) were computed for the models, as well as for RMI, ROMA, and CA125 using established cut-offs. The sensitivity achieved in the original models at these cut-offs is henceforth referred to as target sensitivity. The target sensitivity was then used to find clinically relevant cut-offs for the logistic regression models (GOT 1–3), and specificity was calculated from this cut-off. Moreover, sensitivity for the models was calculated at 75% specificity.

Performance of the logistic regression models was validated using leave-one-out cross validation on the entire patient cohort [35]. This is an iterative procedure where the cohort is split into a training set (*n*-1 samples) and a validation set (1 sample) and the model fitted to the training set. The model is then tested on the validation set and the procedure repeated *N* times, such that every sample is used as the validation set once (Figure 2).

## 3. Results

In the original study, a total of 638 women aged 18–87 were included (Figure 1; Table 1). In the model validation 580 women were included (Benign = 445, EOC = 135). Subgroup analyses were performed on benign vs. early (I + II) and late (III + IV) stage EOC, benign vs. Type I and Type II EOC, benign vs. BOT, benign vs. metastasis and non-epithelial tumors.

### 3.1. Model Performance—Gothenburg Index (GOT)

The clinically used algorithms RMI and ROMA for ovarian cancer diagnostics were modified by the addition of HE4 and TVU, respectively. CA125 alone was compared to an algorithm of the combined markers CA125 + HE4 (Table 2).

#### 3.1.1. Risk of Malignancy Index with the Addition of HE4—GOT-1

When applying the RMI algorithm with a cut-off of 200 to differentiate between benign and ovarian tumors in the patient cohort (*N* = 580), sensitivity was 92% and specificity was 84%. RMI was subsequently combined with HE4 in a logistic regression model used to derive the following formula. The intercept was dropped for simplicity.
(1)GOT–1=0.62×log2(RMI)+1.05×log2(HE4)

Here 1.86 and 2.85 denote the odds ratio of RMI and HE4, respectively. Both predictors were significant at *p* < 0.001. Model goodness of fit was significant at *p* < 0.001, indicating a better fit than the baseline model (with only RMI). A clinically relevant cut-off for GOT-1 was determined by setting sensitivity to 92% (target sensitivity), giving a cut-off of 32 and specificity of 86%. The sensitivity of RMI and GOT-1 were evaluated at specificity 75%, where sensitivity was 97% and 96%, respectively. The AUC for both algorithms was 0.95 (Table 2). With GOT-1, false positives decreased from 70 to 60 women of which 8 tumors were shown to be endometrioma.

#### 3.1.2. CA125 with the Addition of HE4—GOT-2

Biomarker CA125 was used to differentiate between benign and malignant ovarian tumors with cut-off > 35, giving sensitivity 93% and specificity 68%. CA125 was then combined with HE4 using logistic regression, and the model was simplified into the following formula.
(2)GOT–2=0.59×log2(CA125)+1.31×log2(HE4)

Both CA125 and HE4 were significant at *p* < 0.001, with an odds ratio of 1.80 and 3.72, respectively. The goodness of fit test indicated a significantly better fit than the baseline model with only CA125 (*p* < 0.001). A cut-off of 33 was determined for GOT-2 by setting sensitivity to 93%. This resulted in specificity 79%, a substantial improvement over CA125 alone (68%). At specificity 75%, sensitivity was 88% and 93% for CA125 and GOT-2, respectively. The AUC increased from 0.92 to 0.94 (Table 2). There were in total 137 (31%) false positives when using the CA125 marker alone. The major types of histology present were endometrioma (*n* = 48), simple cysts (*n* = 27), serous adenoma (*n* = 20), and teratoma (*n* = 17). False positives were reduced to 86 (19%) women with GOT-2. Again, the endometrioma (*n* = 22) histology was more correctly classified.

#### 3.1.3. Risk of Ovarian Malignancy Algorithm with Addition of Transvaginal Ultrasound—GOT-3

The ROMA algorithm with established cut-offs (Pre-M ≥ 11.4%, Post-M ≥ 29.9%) was used to distinguish between benign and malignant ovarian tumors. In the premenopausal and postmenopausal groups, the sensitivity/specificity were 87%/81% and 93%/77%, respectively (Table 2). We acknowledge that there were only 23 women diagnosed with EOC in the premenopausal group and therefore the data should be interpreted with caution. ROMA was combined with a TVU score in a logistic regression model, which was used to derive the following formula.
(3)GOT–3=7.02×ROMA+0.68×TVU

Both ROMA and TVU were significant at *p* < 0.001, with odds ratio 1123.74 and 1.97, respectively. The test for goodness of fit was significant at *p* < 0.001 against the baseline model with only ROMA. Clinically relevant cut-offs for GOT-3 were found by setting sensitivity to 87% (Pre-M) and 91% (Post-M), giving cut-offs 113 and 344, respectively. This resulted in specificity of 88% for Pre-M women and specificity of 80% for Post-M women, both higher than the corresponding values for ROMA alone (81% and 77%). When GOT-3 was compared to ROMA at specificity 75%, sensitivity increased from 87% to 91% (Pre-M) and from 93% to 94% (Post-M). The AUC value showed a small improvement in Pre-M (increased from 0.93 to 0.94) but not in Post-M (0.94 for both; Table 2).

### 3.2. Subgroup Analyses for GOT-1 and GOT-2 Models

In subgroup analyses of early (I + II) and late (III + IV) stage ovarian tumors, GOT-1 improved specificity from 84% to 86% in early stage tumors and to 90% in late stage tumors (Table 3). GOT-2 improved specificity from 68% to 74% in early stage tumors and to 81% in late stage tumors. AUC improved in GOT-2 early stage 0.84 to 0.88 and late stage 0.96 to 0.98 (Table 3).

In subgroup analyses of Type I and Type II ovarian tumors, GOT-1 improved specificity from 84% to 85% in Type I and to 89% in Type II tumors (Table 4). GOT-2 improved specificity from 68% to 80% in Type I and to 71% in Type II tumors. AUC improved in GOT-2 Type I tumors 0.85 to 0.87 and Type II tumors 0.94 to 0.96 (Table 4).

In subgroup analyses of borderline type tumors, GOT-3 improved specificity for borderline type tumors (BOT) in the premenopausal group from 60% to 70%. No difference was seen in the postmenopausal group. No improvement was seen for GOT-1 and GOT-2. In the subgroup analyses of the metastasis, GOT-1 improved specificity from 44% to 59%. No other improvements were seen.

## 4. Discussion

In the present study, we have evaluated existing, well-known algorithms (RMI and ROMA) and serum biomarkers (CA125 and HE4) in novel combinations in a large cohort of unselected women. We present three new models with increased specificity at maintained sensitivity and better or maintained diagnostic accuracy, GOT-1, GOT-2, and GOT-3.

A variety of serum-based ovarian tumor markers have been identified and several multi-marker tests have been created in an attempt to improve diagnosis and early detection of EOC [9,16,19,26,27,28,31,36,37,38,39]. TVU, serum CA125, RMI, and ROMA are internationally most accepted, and are widely used for differential diagnostics in women with an unspecific pelvic mass. The new models demonstrated in this paper clearly show that with the addition of HE4 to RMI or to CA125, we can reduce the number of patients with false positive results substantially. This means that we may dare to choose a more conservative handling of the adnexal mass in a primary diagnostic operation without the need to refer the patient to overloaded tertiary centers. As a result, surgical menopause can be avoided, less morbidity achieved, as well as an increase in the quality of life [40].

There is a need for studies on how to implement the established algorithms with new biomarkers and, if possible with a TVU score. A few recent studies have addressed this question [26,28,37,38]. HE4 and CA125 in combination with age (Copenhagen index-CPH1) or menopausal status (ROMA) improved sensitivity and specificity in early detection but the specificity is still low, especially in the premenopausal group [3,16,28]. With our ROMA + TVU score model, we could increase specificity from 81% (ROMA alone) to 88% (ROMA + TVU) in the premenopausal group. The GOT-2 combination model CA125 + HE4 increased specificity from 68% (CA125 alone) to 79% (CA125 + HE4), hereby decreasing the number of false positives. This confirms prior knowledge that the addition of HE4 substantially increase specificity and should be used in the differential diagnosis of women with pelvic tumors [2,14,41,42]. HE4 may be affected by several factors other than cancer [15]. For example, HE4 levels in women 18-46 years is constant and has a value around 49–51 pmol/L for this age range. The HE4 level then increases continuously until 86 years to a value of 73 pmol/L [43]. Age is a more precise and simpler data point than menopausal status and therefore is easier to use [28].

Moore et al. demonstrated that the dual marker of CA125 and HE4 improved sensitivity to 76.4% at a set specificity of 75% compared to 43% for CA125 alone in detection of EOC [36]. Later, the same group developed the ROMA score [16]. The possible benefits of adding HE4 to CA125 were confirmed in a single center setting at a tertiary hospital [3] and in the present study with a multicenter setting and an unselected population from both secondary and tertiary hospitals [2]. In the current study, sensitivity increased from 88% (CA125 alone) to 93% (CA125 + HE4) at specificity 75%. An increased sensitivity was also demonstrated in the ROMA+TVU model for both premenopausal (87% to 91%) and postmenopausal (93% to 94%) women. In the present study, RMI alone and in combination with HE4 outperformed ROMA with higher sensitivity (97% and 96% respectively) at 75% specificity but had comparable diagnostic accuracy (0.95 and 0.94 respectively).

In a preoperative setup, it is important to correctly identify a malignancy i.e., aim for high sensitivity of the preoperative tests. The RMI with cut-off > 200 allows women with an advanced disease to be correctly referred to tertiary centers for optimal care and is widely used in countries centralizing ovarian cancer surgery [6,7]. Tests with high specificity are important in the selection of women for treatment at the correct level of care and to avoid unnecessary surgery. In the current study, we can demonstrate that the combination of RMI + HE4 improved specificity at target sensitivity compared to RMI alone (86% and 84% respectively), while AUC did not change (0.95 and 0.95). Our results are in line with a recent and comparable study by Yanaranop et al. who developed a linear transformed model including menopausal status, TVU score, CA12,5 and HE4 named R-OPS [38]. Both models GOT-1 and R-OPS presented higher specificity (86% and 80%) than RMI and ROMA (specificity 75% and 70%, respectively) [38]. The IOTA group reported that RMI was inferior to their LR2 model, simple rules and even subjective assessment (pattern recognition), and they recommend using an expert ultrasound examiner to refer the correct patients to a tertiary center [30]. In the current study, only TVU score according to the RMI model has been used [2,19].

The strengths of this study include the multicenter approach in an unselected population and a large prospective cohort. The inclusion of women has been performed in both secondary and tertiary hospitals. There are a few large (>500 women) cohort studies published [16,27,28], where the majority mainly included women at a single or tertiary center, or were composed of a cohort < 500 women [3,26,37,38,44].

It could be discussed that the TVU examiners used in our study are not experts in gynecologic ultrasound. When performed by an experienced sonographer, TVU has been shown to outperform models such as ROMA and RMI [30]. In the R-OPS study, an experienced single user sonographer performed TVU, which might explain the high sensitivity of 94% of RMI [38]. In a general clinical setting, the possibility for TVU by an experienced sonographer will be limited. We would like to argue that the applicability of the results are strengthen by the fact that the TVU has been performed by a gynecologist with mixed competence, which reflects the reality represented in many health centers.

In spite of the large population used in this study, which is to our knowledge the largest cohort of unselected women with ovarian cysts or pelvic tumors, it is still too small to perform adequate subgroup estimations. The results from Pre-M and Post-M groups should be interpreted with caution as well as the early/late stage ovarian cancer and Type I/Type II ovarian cancer comparisons. Another limitation with our present study is the lack of an external validation. The training and validation is performed with adequate statistical modelling of the available data set but the results would be strengthened if the algorithms could be tested on independent data, but this is not available at the moment. In an independent data-set optimal cut-off with regard to the best sensitivity and specificity could be calculated.

Finding new tumor-associated markers is an ongoing attempt in the research field aiming at enhancing sensitivity, but not at the expense of diminished specificity. The OVA500 [31] initially reported by the multimarker index assay (OVA1) pivotal study [39], which was FDA cleared in 2009 for presurgical risk assessment, analyzed Apolipoprotein A-1, transthyretin, beta-2-microglobulin, transferrin (TRF), and CA125. This improved the sensitivity compared to CA125 alone but at the cost of a high number of false positive cases [31]. The second generation mutlimarker index assay (MIA2G), a validation study of OVA500 analyzing Apolipoprotein A-1, CA125, HE4, FSH, and TRF on a Roche platform, was conducted to improve specificity while maintaining a high sensitivity [27]. CA125, HE4, and menopausal status were recently combined with a multi-marker assay (chitinase-3-like protein (YKL-40), transthyretin, Apolipoprotein A-1, beta-2-microglobulin, transferrin, and lysophosphatidic acid (LPA) [37]. This eight-marker assay improved the diagnostic accuracy but the new model showed no statistically significant improvement compared to ROMA alone (*p*-value = 0.078). Clinical implementation of more expensive assays that are no better than existing ones is not relevant.

## 5. Conclusions

In this multicenter clinical trial in the Western part of Sweden [2], an attempt was made to improve established serum biomarker and TVU-based algorithms. We have modified and validated the use of HE4 and TVU in addition to already established markers and algorithms instead of incorporating expensive new ones. In the new models (GOT-1/GOT-2/GOT-3) created by combining RMI + HE4, CA125 + HE4, and ROMA + TVU, diagnostic accuracy only slightly increased or was the same, even though the models were significantly better when we assessed goodness of fit with the likelihood ratio test. Sensitivity was the same or increased, but the large benefit with the new models was the increased specificity at target sensitivity using recommended cut-offs. The results again highlight the role of HE4 as a complement to CA125, especially when classifying women with benign ovarian cysts. Our new models are superior to RMI, ROMA, and CA125 alone in the differential diagnostics of a pelvic mass.

## Figures and Tables

**Figure 1 jcm-09-00299-f001:**
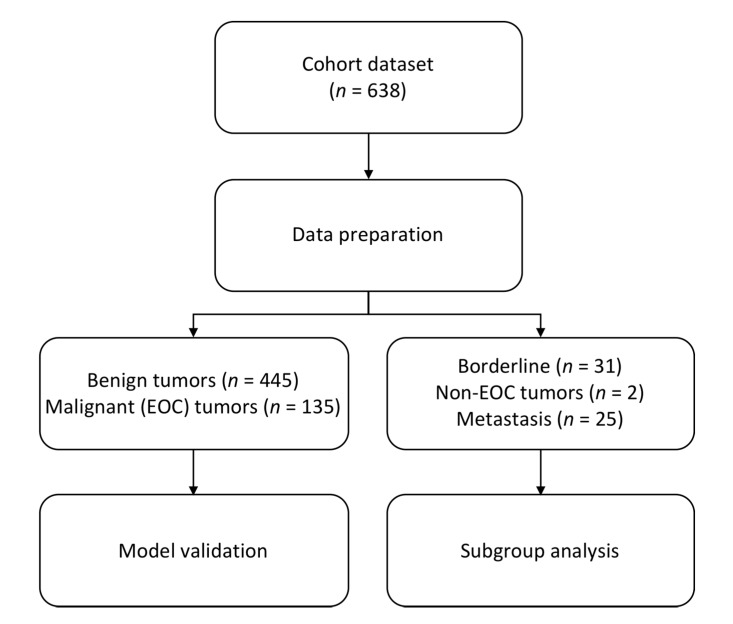
Flowchart of study design and included patients. EOC = epithelial ovarian cancer; *n* = number.

**Figure 2 jcm-09-00299-f002:**
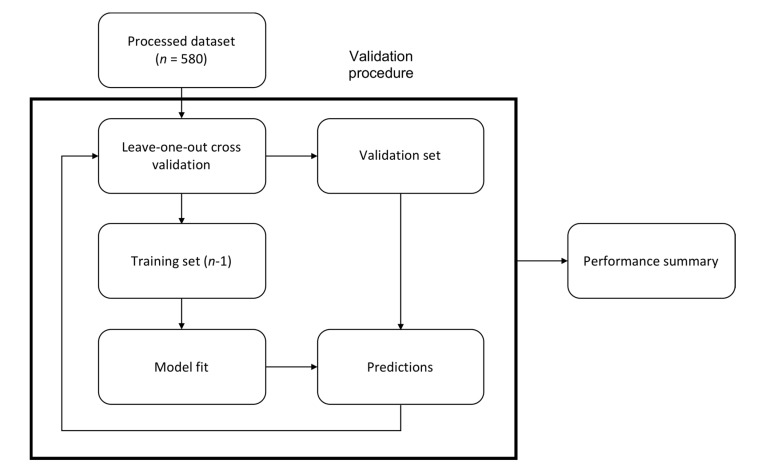
Validation procedure. Regressions were validated by leave-one-out cross validation, where the cohort was repeatedly split into a training set (*n*-1) and a validation set. The training set was used to fit a model and the validation set was used for testing. The procedure was repeated *N* times to calculate the performance of the model.

**Table 1 jcm-09-00299-t001:** Type I and Type II, stage and clinical characteristics.

		Pre-M *n*	Post-M *n*	All *n* (%)
Benign				
age (mean)		38.76	63.6	50.76
Histology *n*				
	Serous	21	76	97 (21.8)
	Mucinous	26	33	59 (13.3)
	Endometrioma	53	7	60 (13.5)
	Simple	64	52	116 (26.1)
	Stromal	3	15	18 (4.0)
	Inflammation	6	8	14 (3.1)
	Teratoma	47	11	58 (13.0)
	Myoma	10	13	23 (5.2)
Total *n* (%)		230	214	445 (69.8)
Borderline				
age (mean)		38.2	63.86	55.58
Histology *n*	Serous	5	13	18 (58.1)
	Mucinous	5	7	12 (38.7)
	Stromal		1	1 (3.2)
Total *n* (%)		10	21	31 (4.9)
Malignant				
age (mean)		44.26	66.46	62.67
Histology *n*	EOC			
	Serous	12	85	97 (71.9)
	Mucinous	4	8	12 (8.9)
	Endometrioid	4	11	15 (11.1)
	Clearcell	3	2	5 (3.7)
	Carcinosarcoma		3	3 (2.2)
	Undifferentiated		2	2 (1.5)
Total *n* (%)		23	112	135 (21.2)
	Non-epithelial OC	1	1	2 (0.3)
	Metastasis	7	18	25 (3.9)
Type I/II	I	9	27	36
	II	14	85	99
Total *n* (%)		23	112	135
FIGO	I	6	30	36
	II	3	13	16
	III	12	56	68
	IV	2	13	15
Total *n* (%)		23	112	135 (21.2)

EOC = epithelial ovarian cancer; FIGO = International Federation of Gynecology and Obstetrics; *n* = number; Pre-M = premenopausal; Post-M = postmenopausal.

**Table 2 jcm-09-00299-t002:** Performance of RMI, CA125, ROMA, and new models GOT-1 (RMI + HE4), GOT-2 (CA125 + HE4) and GOT-3 (ROMA + TVU), comparing benign disease with EOC.

Group (*n*)	Model	*p*-Value	ROC		SN % (75% SP)	SP % (Target SN)
			AUC	95% CI		
Benign (445) vs. EOC (135)	RMI3 (cut-off > 200)	<0.001	0.95	0.93–0.97	97	84
	GOT 1 (RMI + HE4)		0.95	0.93–0.98	96	86
Benign (445) vs. EOC (135)	CA125 (cut-off > 35 U/mL)	<0.001	0.92	0.89–0.94	88	68
	GOT 2 (CA125 + HE4)		0.94	0.92–0.97	93	79
Benign (230) vs. EOC (23) Pre-M	ROMA (cut-off ≥11.4%)	<0.001	0.93	0.86–1.00	87	81
	GOT 3 (ROMA + TVU)		0.94	0.87–1.00	91	88
Benign (215) vs. EOC (112) Post-M	ROMA (cut-off ≥ 29.9%)	<0.001	0.94	0.91–0.96	93	77
	GOT 3 (ROMA + TVU)		0.94	0.91–0.96	94	80

*p*-values calculated with the likelihood ratio test, comparing the regression models to their baseline models. *p*-value < 0.05 was considered statistically significant. Specificity achieved calculated using target sensitivity (Target SN). Target SN GOT-1 = 92%, target SN GOT-2 = 93%, and target SN GOT-3 = 87%/91% (Pre-M/Post-M). AUC = area under the curve; GOT 1 = Gothenburg index 1; GOT 2 = Gothenburg index 2; GOT 3 = Gothenburg index 3; Pre-M = premenopausal; Post-M = postmenopausal; RMI = Risk of malignancy index; ROMA = Risk of malignancy algorithm; ROC = receiver operating characteristics; SN = sensitivity; SP = specificity; TVU = Transvaginal ultrasound.

**Table 3 jcm-09-00299-t003:** Subgroup analyses for validation of performance of RMI, CA125, GOT-1 (RMI + HE4), and GOT-2 (CA125 + HE4) comparing benign disease with EOC in early-and late stage tumors.

Group (*n*)	Model	FIGO I + II				FIGO III + IV			
		ROC		SN% (75% SP)	SP% (Target SN)	ROC		SN% (75% SP)	SP% (Target SN)
		AUC	95% CI			AUC	95% CI		
Benign (445) vs. EOC FIGO I + II (52)/FIGO III + IV (83)	RMI (cut-off < 200)	0.90	0.85–0.94	94	84	0.98	0.97-0.99	99	84
	GOT-1 (RMI + HE4)	0.90	0.86–0.95	92	86	0.98	0.97-1.00	99	90
Benign (445) vs. EOC FIGO I + II (52)/FIGO III + IV (83)	CA125 (cut-off > 35 U/mL)	0.84	0.79–0.90	75	68	0.96	0.94-0.98	96	68
	GOT-2 (CA125 + HE4)	0.88	0.82–0.93	85	74	0.98	0.96-1.00	99	81

Specificity was calculated using target SN. Target SN = target sensitivity of RMI at cut-off >200 was 83% (early stage) and 98% (late stage); target sensitivity of CA125 at cut-off >35 was 85% (early stage) and 98% (late stage). AUC = area under the curve; EOC = epithelial ovarian cancer; FIGO = International Federation of Gynecology and Obstetrics; FIGO I+II = Early stage tumors; FIGO III+IV = EOC late stage tumors; GOT-1 = Gothenburg index 1; GOT-2 = Gothenburg index 2; Pre-M = premenopausal; Post-M = postmenopausal; RMI = risk of malignancy index; ROC = receiver operating characteristics; SN = sensitivity; SP = specificity.

**Table 4 jcm-09-00299-t004:** Subgroup analyses for validation of performance of RMI, CA125, GOT-1 (RMI + HE4), and GOT-2 (CA125 + HE4) comparing benign disease with EOC in dualistic model Type I and Type II tumors.

Group (n)	Model	Type I				Type II			
		ROC		SN% (75% SP)	SP% (Target SN)	ROC		SN% (75% SP)	SP% (Target SN)
		AUC	95% CI			AUC	95% CI		
Benign (445) vs. Type I (36)/Type II (99)	RMI (cut-off <200)	0.89	0.84–0.94	94	84	0.97	0.95–0.98	98	84
	GOT-1 (RMI + HE4)	0.90	0.84–0.95	94	85	0.97	0.95–0.99	97	89
Benign (445) vs. Type I (36)/Type II (99)	CA125 (cut-off >35 U/mL)	0.85	0.79–0.91	75	68	0.94	0.91–0.97	93	68
	GOT-2 (CA125 + HE4)	0.87	0.82–0.93	83	80	0.96	0.94–0.99	96	71

Specificity was calculated using target SN. Target SN = target sensitivity of RMI at cut-off >200 was 83% (type I) and 95% (type II); target sensitivity of CA125 at cut-off >35 was 83% (type I) and 96% (type II). AUC = area under the curve; EOC = epithelial ovarian cancer; GOT-1 = Gothenburg index 1; GOT-2 = Gothenburg index 2; Pre-M = premenopausal; Post-M = postmenopausal; ROC = receiver operating characteristics; SN = sensitivity; SP = specificity; Type I = low-grade tumors; Type II = high-grade tumors.

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
