# Peer review of "Increased Diagnostic Accuracy of Adnexal Tumors with A Combination of Established Algorithms and Biomarkers"

_jcm, 2020, doi:10.3390/jcm9020299_

Round 1

Reviewer 1 Report

The present study aims to evaluate the predicitve capacity of a new algorithm on diagnosing ovarian cancer. In this attempt, a prospective mulitcenter study was initiated. The research question is novel and of interest to the journal's readers. The study design appears appropriate to answer the posed question, though subgroup analyses have to be interpreted with caution.

There are just some points that deserve consideration:

The novelty of the presented work is closely associated with the inclusion of Human Epididymis Protein 4 (HE4) into pre-existing algorithms for the detection of ovarian cancer. I therefore encourage the authors do provide some details on this marker in their introduction. Additionally, it would be of interest for the readers to Point out conditions of false positives.

The main drawback of this study is the lack of a control group with HE4 measurements. It would be helpful if this could be added. Even a small sample size would be helpful.

Author Response

Please se the attachement.

Reviewer 2 Report

It is an interesting and well written article. Despite the complexity of juggling markers alone and algorithms, the manuscript is fairly easy to read.
I would have 3 major revision to make:
- sensitivity / specificity
p2 l68: I agree on the choice of specificity and reduction of FP. Explain just why you are not choosing sensitivity (which is played by ultrasound or imaging). On the other hand in these cases, why do you choose a "target sensitivity" (p6 l170, p6 l184, p7 l205)? Why don't you choose the best specificity for your cut-off, since your population is already targeted?
- format modification
p3 the flowchart and table 1 must be part and must be discussed in the results part and not material and methods
- some interesting articles are missing in the bibliography
 . Human epididymis protein 4: factors of variation.Clin Chim Acta. 2015 Jan 1; 438: 171-7. doi: 10.1016 / j.cca.2014.08.020. Epub 2014 Aug 27. To quote, because specifies the factors of variations of HE4 (as done with CA125)
 . Biomarkers and algorithms for diagnosis of ovarian cancer: CA125, HE4, RMI and ROMA, a review.J Ovarian Res. 2019 Mar 27; 12 (1): 28. doi: 10.1186 / s13048-019-0503-7. To quote, because it is the most recent review on the subject
 . Serum HE4 and diagnosis of ovarian cancer in postmenopausal women with adnexal masses. Am J Obstet Gynecol. 2019 Jul 24. pii: S0002-9378 (19) 30936-6. doi: 10.1016 / j.ajog.2019.07.031. [Epub ahead of print]. To quote because argues what you also find according to the menopausal statute.
 . Efficacy of HE4, CA125, Risk of Malignancy Index and Risk of Ovarian Malignancy Index to Detect Ovarian Cancer in Women with Presumed Benign Ovarian Tumours: A Prospective, Multicentre Trial. J Clin Med. 2019 Oct 25; 8 (11). pii: E1784. doi: 10.3390 / jcm8111784. To quote, because also opens on your conclusion and supposed benign cysts.

The other minor comments:
- p1 l19-20 and p4 l116: your cancer prevalence is high among your population on which you apply your model (25%). Perhaps cite this as a bias or a limit or conversely clarify that your initial selection is very reliable.
- p1 l32-33: the sentence "complete staging ..." is not useful
- p1 l39-40: I don't see the point of talking about debulking surgery here
- p2 l53: space between [13].the TVU
- p3 l100: describe pre-M and post-M in the text
- p9 l276-285: this paragraph is in my opinion not very useful because it speaks of the sensitivity and not of the specificity
- p10 l324-325: this is normal if we favor sensitivity, we often increase FP, hence the interest of these markers and algorithms
